# The Obverse/Reverse Pavilion: An Example of a Form-Finding Design of Temporary, Low-Cost, and Eco-Friendly Structure

**Jerzy F. Łątka [1],\* and Michał Święciak [2]**

1   Department of Architecture and Visual Arts, Faculty of Architecture,
    Wroclaw University of Science and Technology, 50-317 Wroclaw, Poland
2   Super Architektura Michał Święciak, 86-300 Grudziądz, Poland; SuperArchitektura@gmail.com
\*   Correspondence: jerzy.latka@pwr.edu.pl

**Abstract:** Temporary pavilions play an important role as experimental fields for architects, designers, and engineers, in addition to providing exhibition spaces. Novel structural and formal solutions applied in pavilions also can give them an unusual appearance that attracts the eyesight of spectators. In this article, the authors explore the possibility of combining structural novelty, visual attractiveness, and low cost in the design and construction of a temporary pavilion. For that purpose, an innovative structural system and design approach was applied, i.e., a membrane structure was designed in Rhino and Grasshopper environments with the use of the Kiwi!3D IsoGeometric analysis tool. The designed pavilion, named Obverse/Reverse, was built in Opole, Poland, for the occasion of World Architecture Day in July 2019. The design and the construction were performed by the authors in cooperation with students belonging to the Humanization of Urban Environment organization from the Faculty of Architecture Wroclaw University of Science and Technology. The resultant pavilion proved the potential of obtaining a low-budget but visually attractive architectural solution with the adaption of parametrical design tools and some scientific background with innovative structural systems.

**Keywords:** parametric design; paper in architecture; temporary architecture; pop-up structures; membrane structures; isogeometric analysis; fabrication





## 1. Introduction

Today, more and more events such as festivals, trades, and exhibitions require individual spatial design that should attract eyesight in order to draw the attention of passers-by. Often, this function is fulfilled by temporary pavilions. Although an event may last for a very limited time, small architecture structures are usually built from traditional materials and in a simplified manner, i.e., rectilinear forms, resulting from limitations of using conventional CAD tools.

The construction of temporary pavilions makes it possible to highlight the technological, cultural, or artistic values and advancements of a presenting person, social group, or even whole nations during events, celebrations, or fairs. The most famous events allowing for such displays are the World Fairs (Expos) during which nations exhibit their cultural, scientific, and technological legacy and advancements. Those expositions take place in pavilions designed especially for these occasions by architects, usually selected through a competition or to honor their current achievements. Temporary pavilions, originally intended for housing the actual exhibition, have become over the years the subject of these exhibitions in their own right. Among the 'temporary' pavilions built on the occasion of World Fairs, some were made using breakthrough technological solutions and they are still standing today (e.g., Portugal's Pavilion—Figure 1). Pritzker award-winning architect Frei Otto was the author of Germany's Pavilion during the World Fair in Montreal in 1967, which disseminated the use of membrane structures. Today, temporary pavilions that display cultural heritage, feature technological advancement, or constitute artistic manifests occur also during events of more minor importance than World Fairs. Every year

since 2000, the Serpentine Galleries in London invite a world-famous architect to design and build a pavilion that can attract visitors just by its appearance [1].

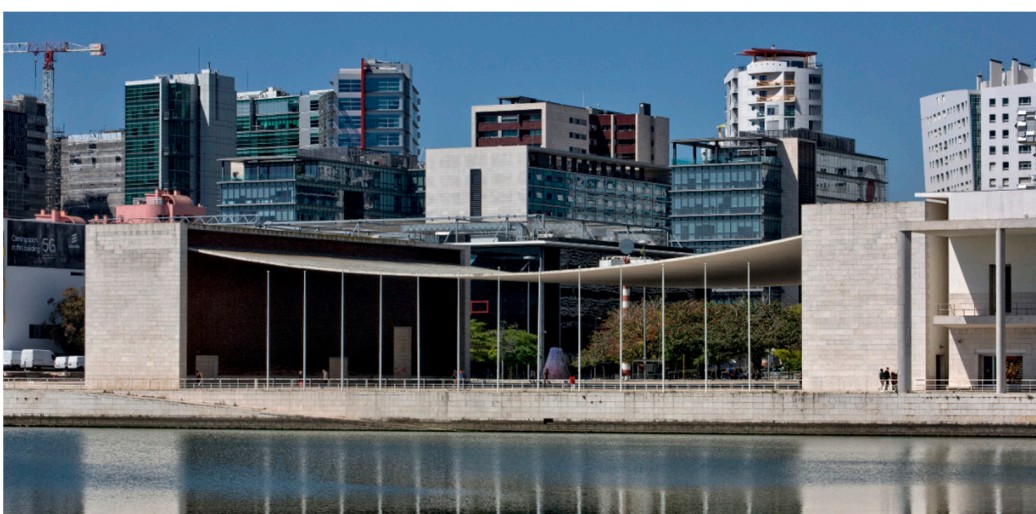

**Figure 1.** Portugal Pavilion for Expo Lisbon '98—architect Alvaro Siza.

Pavilions attract eyesight as something "which has not been seen before", especially as built objects which stand and can even be touched, contrary to pictures which can be arbitrarily altered and depict unreal and impossible structures. In such sense, temporary pavilions have a greater impact on spectators. In order to stand out from permanent structures and other pavilions, novel and innovative geometrical and structural solutions are willingly used. Tight cooperation between architects and engineers is required in order to achieve solutions characterized by extreme properties such as volume-to-mass ratio or constructional section area to span. The example of the TRADA Pavilion showed the potential of parametric design software in designing a small-scale pavilion (Figure 2a). This 4 × 6 × 4 m pavilion was designed and built by Ramboll Computational Design in 2012 with the use of a zero-length spring funicular form-finding approach combined with a planar polygon discretization. The design method resulted in a curved form created from planar elements. The structure was made using 152.15 mm thick plywood panels connected to each other by means of hinges [2].

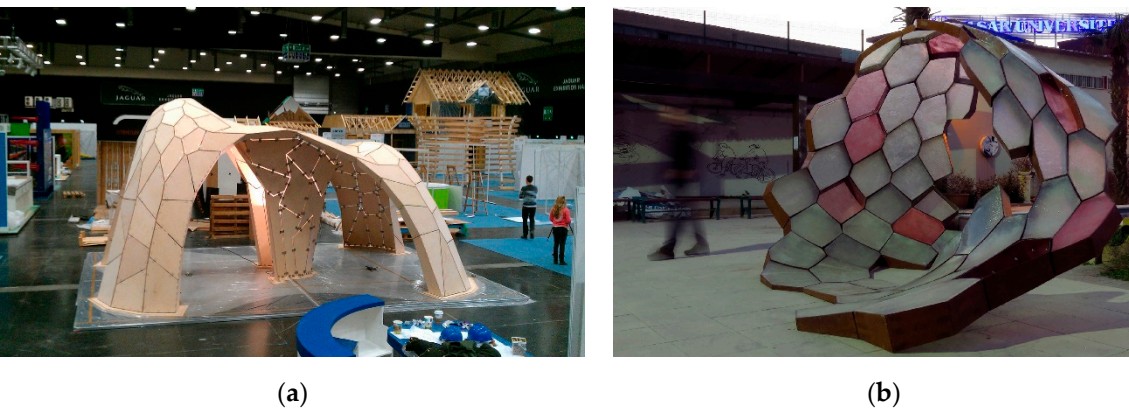

(**a**)          (**b**)

**Figure 2.** Temporary pavilions: (**a**) the TRADA Pavilion, 2012, reproduced with permission from John Harding; (**b**) TU Delft and Yasar University Pavilion, 2015, reproduced with permission from Serdar Asut.

The pavilion created in 2015 by students from TU Delft and Yasar University in Izmir, supervised by Serdar Aşut, Winfried Meijer, Peter Eigenraam, and Thijs Welman from TU Delft, Bilge Göktoğan from YU, and Mark Giraud from Polkima Plastic Solutions,

showed the possibility of using parametric software with the use of the human numeric control (HNC) production method (Figure 2b). HNC, in contrast to computer numeric control (CNC), requires interaction between machine and human. The pavilion's panel elements were designed with the use of Grasshopper and produced with FlexiMold, a device developed at TU Delft. FlexiMold is a formwork with dimensions of 70 by 70 cm constructed using 49 steel rods that can be manually positioned in a vertical direction and covered with flexible sheet. The user receives data from the digital model and positions the rods, thus achieving the desired curvature when casting the panel. The pavilion in the form of a mobius stripe had dimensions of 4 m in length and 2.3 m in height, and it was composed of 59 unique double-curved panels made of fiberglass flanged with MDF boards [3].

The Packed Pavilion created by students from ETH Zurich under the supervision of Tom Pawlofsky is an example of the use of low-tech material, i.e., corrugated cardboard, in a temporary structure (Figure 3a). The pavilion consisted of 409 truncated cones connected to each other by means of zip ties. Each of the cones was made of 28 layers of corrugated cardboard, which were cut and laminated with the use of a computer-controlled machine. The cones with different diameter fitted into one another, which reduced the amount of material and minimized the volume for transportation. The cones were protected against water and humidity by shrink foil. The Packed Pavilion, which was produced in Switzerland and shipped to the Shanghai Expo in 2010, aimed to present how CAAD can be exploited for the design, production, and optimization of material use, the production process, and logistics [4].

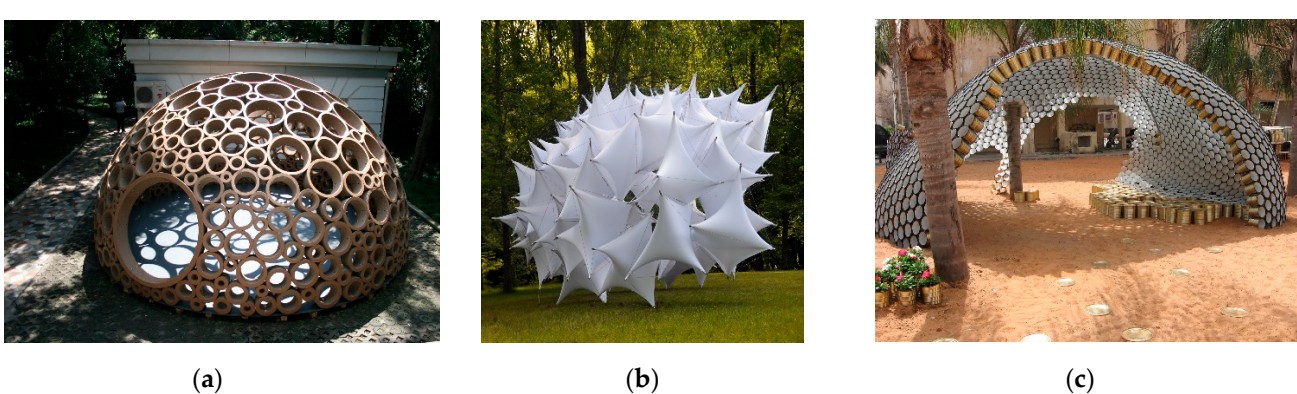

| (a) | (b) | (c) |

**Figure 3.** Temporary pavilions: (**a**) the Packed Pavilion, 2010, reproduced with permission from Dominik Zausinger; (**b**) the Underwood Pavilion, 2014, reproduced with permission from Gernot Riether; (**c**) the Can Pavilion, 2008, reproduced with permission from Lihi Ein-Gedi Davidovich.

The Underwood Pavilion was an experimental tensegrity structure created in 2014 by the students of Ball State University under the supervision of Gernot Riether, Andrew J. Wit, and Steven T. Putt (Figure 3b). The structure was developed by means of Rhinoceros 3d, Grasshopper, Galapagos, and Kangaroo software, which allowed for a graphic representation of a complex structural system, including the material's physical properties. The structure was developed in a feedback loop; the ideas were tested using a physical model, which was then confirmed using a parametrically designed object, before being subsequently proven as a physical prototype. The pavilion consisted of tensegrity modules whose size varied from 90 × 90 × 90 cm to 120 × 120 × 120 cm. The modules were made of aluminum pipes and steel cables, additionally dressed with elastane, i.e., a stretchable membrane [5].

An interesting example of the use of low-cost materials was the Can Pavilion. The pavilion was designed and built by Lihi Ein Gedi Davidovich, Roee Chido, and Galit Khvous Fleische, students of architecture from Tel Aviv University, for the occasion of the 2008 Urban Landscape Architecture Biennale in Bat Yam, Israel (Figure 3c). As the main theme of biennale was 'hospitality', the authors proposed an innovative structure with

the use of low-tech and local material—soup cans. The pavilion had an open form with a seating area in the shade [6].

The aim of this research was to prove the thesis that, even with a low budget and without specialized engineers and contractors, it is possible to design and build a spectacular pavilion with the use of commonly available parametric design software tools such as Rhinoceros3D, assisted by plugins Grasshopper and Kiwi!3D. The use of low-tech materials allows for the creation of an eco-friendly and low-cost pavilion without compromising the quality of its form, while a parametrically designed membrane structure can become an eye-catcher in urban space.

The aim was derived from the following research question: how can parametric design software be implemented in order to create an attractive, i.e., eye-caching temporary pavilion with a limited budget and with the use of eco-friendly materials?

The main research question was broken into several sub-questions, which are answered in the subsequent paragraphs of this article.

(1)　What is a pop-up structure in an urban context?
(2)　Which low-cost and eco-friendly materials can be used for temporary structures?
(3)　What is form-finding design and how it can be implemented with the use of parametric design software in the process of pavilion creation?
(4)　How can the form of the pavilion be elaborated and produced with the use of parametric design software?
(5)　What is the final result of the design and development process?

In order to answer these questions, a certain research, design, and development methodology was chosen. To begin, a review of the examples of previously created pavilions of similar scale was conducted. The typology, materials, shape, design, and production methods were analyzed.

Next, the design methodology was adopted. It involved setting the design criteria, research on the pop-up phenomenon, as well as its role, characteristics, and meaning in the contemporary urban space, and the abstract formation of the pavilion in relation to its semantic layer.

The material research was focused on paper- and timber-based products, as the most common, low-cost, and low-tech solutions. Subsequently, research on the membrane material was conducted.

The design development phase consisted of research on form-finding methods for the outer skin membrane and their implementation in the design and production process with the use of parametric design software.

The methodology described in the article was adopted to design and build a temporary pavilion with external membrane skin. We employed methods of isogeometric analysis for form-finding, geodesic curves using a patterning and flattening technique in order to obtain patterning of the external membrane skin of pavilion. The employed methods allow preserving a consistent type of NURBS geometry in the pavilion's model throughout all stages of design development (initial modeling, form-finding, patterning, and flattening), which is beneficial for complex geometries with high curvatures [7]. However, improving any of the involved computational methods was outside of the scope of this research.

Our contribution is the combination of these methods in order to obtain a low-cost pavilion while carrying out the design process within a single software environment (Rhinoceros 3D with available plugins).

Subsequently the final object was prefabricated and constructed, and it was observed in natural conditions for 6 months.

The last step was a revision of the research aim.

With regard to the possibility of building the pavilion as a low-cost structure, access to the software was also an important criterion for us; hence, a decision was made to implement the project within one generally available design environment, such as Rhinoceros 3D with additional plugins.

## 2. Interaction between Public Space and Its Users

In a contemporary urban environment, public spaces play a unique role as common areas, where the interaction among their users is interwoven with the interaction between the users and the place itself. These public spaces both create stimuli and are affected and created by other stimuli.

A public space can influence its users' emotions, perception, and sense of the space, as well as physiological comfort. The reception of the space is influenced by sensory systems which can be divided into visual, auditory, basic orientation, haptic, and taste/smell systems [8]. Creations of architects and designers can influence almost all of those sensory systems (or all of them if, for example, a designed object attracts visitors with its scent, i.e., a coffee shop); however, the most influential are the visual and auditory systems (so-called 'far-space' stimuli), as well as haptic ones ('close-space' stimuli) [9].

Public spaces in urban areas are infiltrated by information. The amount of information creates chaos, which in fact becomes disinformation, where every single player in the public market tries to get the attention of a potential visitor. The dominating sensory system is a visual one; however, visual perception can be amplified by the psychophysiological component and other sensory system experiences [10]. This multilayered sensual composition creates a narration about the place and events that occur in architectural scenography.

In public space design, the complementary order of space, movement, and events should be considered simultaneously. The space can be treated as a stage where architectural events take place while the observer is in motion. The conditions, such as users' emotions, perception, and sense of the space, influenced by the legible, narrative design, are key to the successful design of an architectural object in public spaces, especially when its role is to trigger someone's attention by being positively received.

## 3. Pop-Up Structures

Architecture has a significant impact on the cultural regeneration of the city. The realization of a spectacular public building that draws people's attention brings popularity and creates a space for cultural events, thus boosting the economy; this phenomenon already has its name in academic and architectural language—"wow-architecture" or the "Bilbao effect" (after the great success in the development of the Spanish city of Bilbao thanks to the realization of Guggenheim Museum designed by Frank Gehry in the mid-1990 s) [11]. These visionary buildings bring new qualities in neighboring areas, re-establish local identity, and break through traditional barriers.

However, wow-architecture created by famous architects might be a risky investment. As Lähdesmäki states, these big and large investments are not flexible to the changing demands and strategies for cultural regeneration. The other solution for architectural and cultural regeneration might be the use of temporary architectural structures for cultural, commercial, or leisurely use [12]. These event-oriented architectural installations can create spaces or landmarks for cultural activities. They can also achieve the wow-effect, although on a smaller scale and with a much smaller budget involved. Therefore, solutions that bring new and surprising qualities to the existing (i.e., known) context of public spaces are temporary structures or so-called "pop-up architecture".

To pop up means to appear or happen, especially suddenly or unexpectedly; in architectural language, a pop-up refers to places such as shops, restaurants, exhibitions, or pavilions that operate only for a short period of time, when it is likely to get a lot of customers [13].

Although pop-up structures have been known since ancient times, when temporary timber structures had the form of theaters and festival or public games areas, it is since the beginning of the 21st century that they received substantial attention [14]. The popularity of pop-up structures arises from the fact that the urban fabric is fulfilled with permanent structures, while the temporary nature of pop-up structures provides an opportunity for testing unconventional solutions and experimenting in a public space with limited financial

and legislative issues [15]. They are structures or spaces that are built quickly and are intended for temporary use, while fulfilling users' functional and aesthetical needs [16].

Pop-up structures can play the role of a temporary pushpin in urban aquapuncture, creating new quality in existing spaces and focusing the users' attention on the sense of "being" in a particular place [17]. Temporary objects can become catalysts of changes in the existing environment. The example of Shigeru Ban's project of Nomadic Museum shows how a temporary structure made out of shipping containers and paper tubes brought attention to the abandoned Pier 5 on the Hudson River in New York [18].

Temporary architecture is characterized by simplified structural systems and a functional layout. Due to the mobility and lightweight requirements, it is most often a single-story structure. The provisional and experimental characteristics of these objects create an opportunity for the implementation of new technologies and materials, which are often derived from different disciplines and industries.

Pop-up structures in the urban layout are often stylish urban furniture with the use of innovative low-ecological-footprint materials. The materials that temporary objects are made of have an important role in their perception. Low-tech materials complemented by advanced technologies from other industries bring new values such as symbolism, emotions, originality, and environmental concern.

The sensory (visual, haptic) attractiveness of pop-up structures can be achieved by materials used for both construction and architecture. The latter can introduce content and ideas related to the existing social and spatial context of an urban space.

## 4. International Day of Architecture in Opole—A Case Study for Pop-Up Pavilions

Since 2016, the Opole city branch of the Chamber of Polish Architects has organized attractions and events on the occasion of World Architecture Day, established to remember the foundation of the International Union of Architects on 1 July. The all-day-long event focuses on the promotional action of the profession of an architect and their role in society, exhibitions, presentations, workshops, and discussions. At the same time, a temporary pavilion is built every year. The pavilion is prefabricated and then assembled on site. The assembly process is a performance that draws attention and, as such, is planned as part of World Architecture Day. The size of the pavilion is approximately 5 × 5 × 5 m. Every year, the form of the pavilion responds to certain aspects of architectural creation.

In 2016, architect Dawid Rószczka designed and built the Wood and Shadow Pavilion (Figure 4). This cubic form was made out of plywood slender elements that were slid to each other. The whole structure was mounted without the use of any screws or nails. As such, it was practical and easy to assemble and disassemble the pavilion and transport it between places. The voids created by the orthogonal divisions of the structure allowed light to go through and visually change the object during the day, as well as by means of night backlight. When viewed frontally, the pavilion seemed a delicate, lightweight structure; however, when observed at some angles, the thin plywood panels created the impression of a full wall that resembled traditional Japanese window shutters, giving an oriental touch to the structure, which responded to the lectures and exhibitions about Japanese architecture presented during the event. The pavilion was first built at Bastion of St. Jadwiga in the city of Nysa, before being dismantled and rebuilt in the main square of Opole.

One year later, the World Architecture Day in Opole was focused inter alia on the activation of public spaces. Several lectures, discussions, and exhibitions concerned this topic. The pavilion accompanying the event was designed by Dawid Rószczka, Kamila Wilk, and Łukasz Kościuk (Figure 5). The Architectonic Sculpture pavilion was composed of 14 frames with M-like forms made out of timber planks. This tunnel-shaped pavilion was covered with an interlining material membrane in order to diffuse the LED illumination from below. The pavilion was located at the Plac Wolnosci (Square of Freedom) in Opole and served as a needle in the acupuncture of the urban environment. Due to its size and shape, it attracted the attention of passers-by attending other events organized within the

World Architecture Day. Similarly to the year before, the pavilion was first built in Nysa and then transported to Opole.

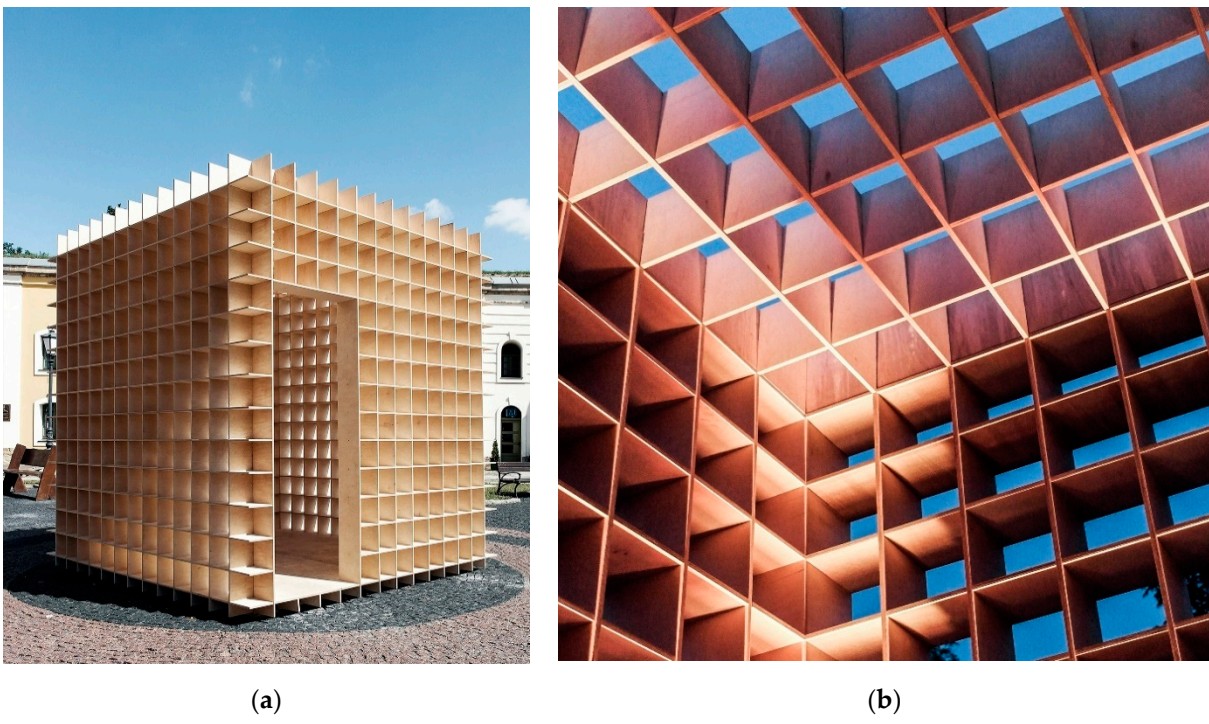

(**a**)          (**b**)

**Figure 4.** Wood and Shadow Pavilion, 2016: (**a**) general view of the pavilion; (**b**) interior of the Wood and Shadow Pavilion, 2016, reproduced with permission from the Chamber of Polish Architects branch in Opole.

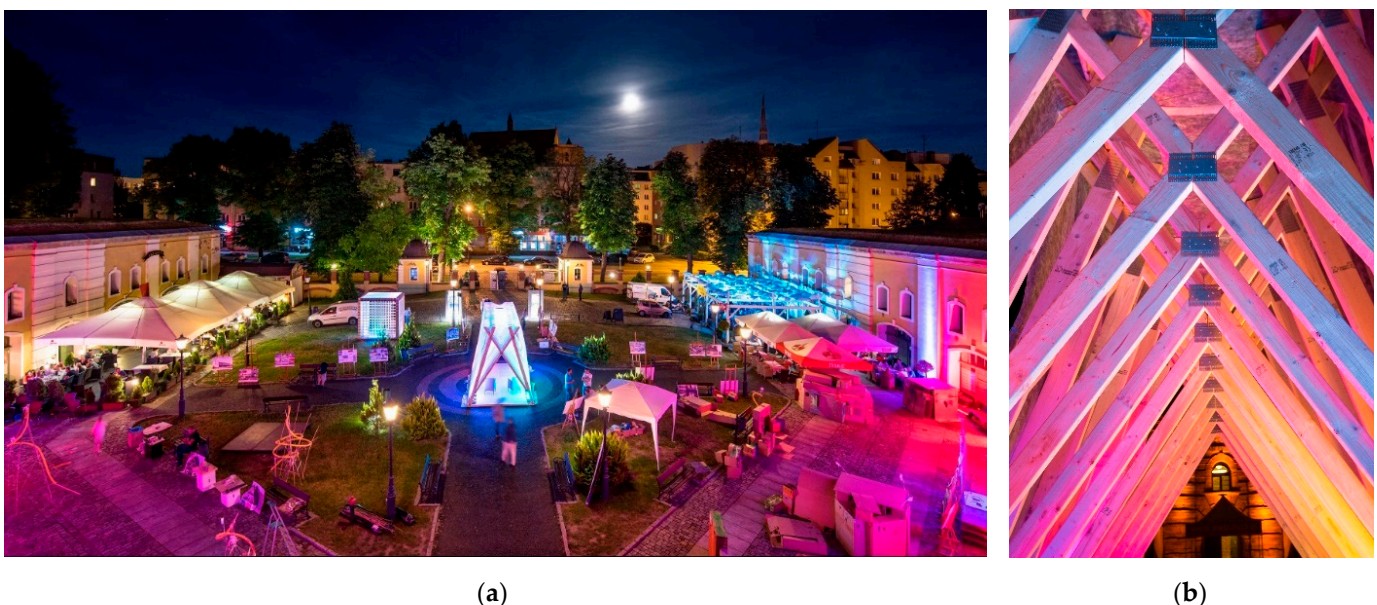

(**a**)          (**b**)

**Figure 5.** The Architectonic Sculpture Pavilion, 2017: (**a**) view at the Pavilion erected in Nysa, photo by Maciej Zych Photography; (**b**) detailed view of the pavilion, reproduced with permission from the Chamber of Polish Architects branch in Opole.

The theme of the World Architecture Day in 2018 was "Architecture and Water". For this occasion, the pavilion was designed by Wiesław Półchłopek and Marcin Zdanowicz (Figure 6). The structure had a rectangular shape. The pavilion was made out of a timber frame clad with plywood plates. The plates had irregular shapes with gaps between, which brought about a dynamic perception assisted by daytime and evening light. The special

feature was represented by three waterfalls created inside of the pavilion. This feature was very appreciated, especially by kids playing around, and it also brought refreshment into the air on hot summer days. Everyone who wanted to cross the pavilion had to take a special route to not get wet, providing a straightforward connection between water and architecture. In the evening, the pavilion was lit with LEDs which changed the colours and, thus, the perception of the structure.

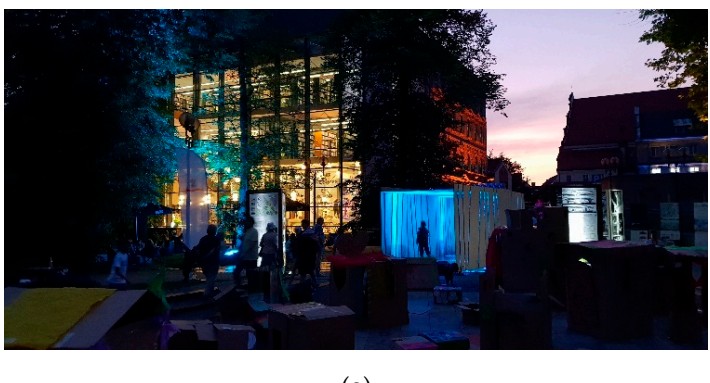 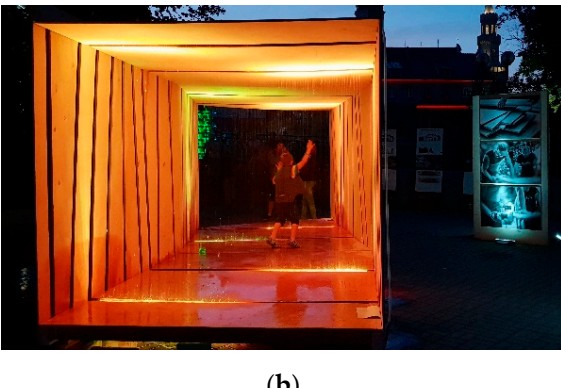

(**a**)  (**b**)

**Figure 6.** Water Pavilion, 2018: (**a**) the view from the at Plac Wolności; (**b**) water installation in the pavilion, reproduced with permission from Chamber of Polish Architects branch in Opole.

## 5. Obverse/Reverse Pavilion—Concept, Materials, and Design Methods

In the year 2019, the theme of the World Architecture Day in Opole concerned ecology and architecture. The coauthor of this article was commissioned by the Chamber of Architecture branch in Opole to design and construct, together with the students from Wroclaw University of Science and Technology, a temporary pavilion that would suit the topic.

The aim of the project was to create a pro-ecological, temporary structure, which would serve as an eye-catcher in the urban public space and attract the attention of passers-by on several levels of interpretation.

The plan was for the pavilion to be exhibited for a few days at Plac Wolności (Square of Freedom) in the city center of Opole, before being moved to another location—a square next to the Contemporary Art Gallery in Opole, where it could bring new value as an urban sculpture.

The creation of this project, aimed at meeting the demands of a pro-ecological characteristic for this temporary object in an urban area, was a goal set by the authors during the design and development process. The form, materials, and structural feasibility were the main design objectives set out at the beginning of the process. It was obvious that the pavilion, similarly to the previous ones, should have an attractive and surprising form, which would draw attention of the passers-by. Therefore, other design criteria aimed at meeting the requirements for pop-up objects in the urban environment were defined as follows:

- the object should have some meaning included in its form, which should be narrative, i.e., tell some story or have a symbolic code incorporated,
- the pavilion should be an eye-catcher, i.e., a surprising, unobvious form which stands out from the urban background and triggers someone's attention in a positive manner,
- the pavilion should influence at least the visual and haptic sensory systems,
- the shape of the pavilion should be unusual and arouse curiosity, along with a soft and friendly (relaxing) but stimulating (not boring) form,
- the form should create some kind of narration (coming closer, going through),
- the form should be inclusive (accessible, open, accompanied by chairs),

- it should have the potential to be a pushpin in the urban area with some sort of wow-effect (with respect to the small scale and limited budget of the object), especially for its later localization next to the Contemporary Art Gallery.

As the main role of the World Architecture Day is to highlight the profession of an architect to a wide audience, the formal goal of this design concerned its symbolism. The decision was made to create a pavilion which referred to the double-sided character of the architectural profession. On the one hand, an architect is an engineer and a rationalist who operates within the constraints of legal, technical and budgetary conditions, who has to face the demands of investors and future users, and who often seeks solutions to complex and accumulated structural, legal, functional, and semantic difficulties. On the other hand, an architect is an artist and a creator who crosses their own boundaries and is often not afraid to dream and think big. The personal development of an architect, even if not seen by the others, is a constant process in a life-long career. Lastly, an architect is a person with many fields of specialization, harboring knowledge in the fields of art, engineering, ergonomic, economy, psychology, sociology, and history [19].

This tension between the rational and romantic sides of the profession of an architect was represented by the form of the pavilion. From the inside, the pavilion took the form of an archetypical house, with straight lines and a double-pitched roof. However, from the outside, this form grew into an unpredicted, soft shape, which expanded outward from the regular, i.e., rational, lines into curved, i.e., creative, peaks. The mind and spirit of an architect grows out of the box as their way of thinking should. These obverse and reverse sides of architectural practice were underlined by the colors. It is commonly known that architects wear black [20]; therefore, this color was used to cover the walls, ceilings, and floors of the house-shaped interior, to enhance this stereotype. On the other hand, the blob-like part was white as a symbol of the freedom of an architect's mind, which can grow unabashedly. Perversely, the superficial perception of the architect was directed inward, while the part that symbolizes their internal development and creativity was directed outward, referring to the direct connection between the architect's creation and the surrounding world.

The final form of the Obverse/Reverse Pavilion was developed in several steps, as shown in Figure 7. First, the traditional shape of a house that formed the interior of the pavilion was skewed. As both ends were bigger than the middle, the pavilion featured an open, inviting form. Next, the longitudinal shape was broken in the middle to reduce the tunnel-like impression and increase the visibility of the pavilion from many directions. The substructure of the pavilion was divided into five frames, which formed the inner house-like shape. This form was covered with a membrane—an outer skin, which in its parametric design achieved an organic, soft shape. The membrane was positioned and stretched over the tubes that were fixed to the frames.

As the World Architecture Day in Opole in 2019 referred to ecology and architecture, it was decided that the main structure of the pavilion would be made to the greatest extent from paper-based products.

*5.1. Paper in Architecture*

Paper is a material of natural origin. Its main component is a cellulose fiber, which is the most common natural polymer on the globe, and its resources are considered to be inexhaustible [21]. The first attempts at the use of paper in architecture were recorded in ancient China and Japan, whereas European attempts started in the second half of the 19th century [22]. The contemporary era of paper architecture began in the mid-1980s when Japanese architect Shigeru Ban, for the first time, used paper tubes as a structural element in an architectural object [23]. Since that time, there has been a large body of research conducted at several universities such as TU Delft, ETH Zurich, Wroclaw University of Science and Technology, and TU Darmstadt. Additional knowledge was gained following the execution of buildings with paper-based structural elements [18,24,25].

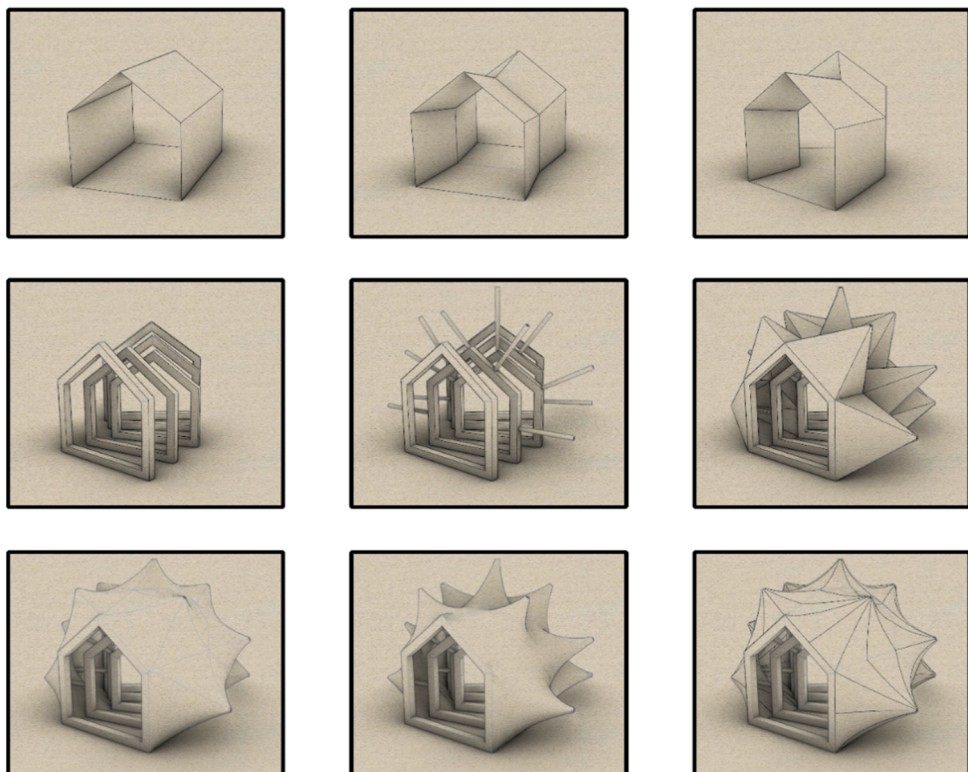

**Figure 7.** Development of the Obverse/Reverse Pavilion.

There are several mass-produced paper-based products which are employed as structural elements of buildings. Paper tubes and full cardboard L- and U-shapes can serve as rod elements, while honeycomb cardboard panels and corrugated cardboard can serve as planar elements. However, paper-based products are often incorporated with other materials for strengthening, protection, or improvement of the properties of paper-based building elements [26].

Paper is fragile when it comes into contact with water. Upon becoming wet, the bonds between cellulose fibers loosen and paper transforms into pulp once again. This feature allows for the recycling the paper elements up to six times; however, each time, some new fibers are added to the batch to strengthen the recycled material [27]. There are several methods used to protect paper elements against water and moisture. They can be coated, painted with various varnishes, covered with foils or other watertight materials, or left natural but placed in such a way so as to minimize their contact with the external environment.

### 5.2. Computational Design of the External Skin

The duality of the pavilion also spoke to relationships between architecture and engineering on two particular levels. The first one was revealed as the pure idea of a building, i.e., a traditional section shape with walls and roof slopes, clearly marked and cut in space, whereas the second, the engineering level, had no explicit shape that was preconceived by the architect. It was expressed on the external side of the pavilion. Its shape was vaguely specified by the interior conditions—the core idea. The external skin of the pavilion is a tensile membrane. Both levels together created an idea dressed up in a structural skin.

The external membrane of the pavilion was designed, form-found, and prepared for fabrication with the use of Rhinoceros 3D and the Kiwi!3D Isogeometric Analysis plugin [28]. Despite Rhino3D not being originally devised as a membrane designing tool, its versatility and add-on resources allow for solving increasingly demanding engineering tasks.

A standard method for describing double-curved surfaces in Rhino 3D, such as membranes, i.e., the mathematical concept of nonuniform rational B-spline (NURBS) surfaces, was used, whereas, in the field of computer-aided engineering, discrete descriptions of forms (meshes) were used for a finite element analysis. Particularly in the context of digital form-finding, a continuously curved form cannot be reliably obtained from a previously discretized (meshed) simulation model. The discretization itself can also be a source of form-finding and calculation errors. It should also be emphasized at this point that Kangaroo3d, shipped with Rhino 3D, is not a finite element method analysis tool, but an interactive physical and constraint solver.

State-of-the-art membrane designing tools based on discretized models are highly specialized and usually offer abilities to analyze and include in designed structures factors such as intrinsic, anisotropic material properties and imperfections, complying with the prestressing forces and elastic deformations of the fabric [29]. These factors should be taken into account for permanent buildings that, during their lifetime, will be subject to varying forces such as wind loads, thermal deformations, and material fatigue, in order to prevent membranes from wrinkling, tearing, or flapping. Taking these factors into consideration also requires the usage of advanced, multilayered membrane fabrics capable of withstanding atmospheric influence without deterioration of their structural properties [30]. Although very lightweight, thin, and efficient, membrane structures pose a challenge in both the design and the production steps.

Computational methods of form-finding are generally divided into discrete and continuous methods, among which the isogeometric analysis (isogeometric B-Rep analysis) [31] and natural force density method [32] were distinct from our research point of view, since both methods are available through Grasshopper addons, i.e., Kiwi!3D and BATS [33]. Isogeometric analysis and the Kiwi!3D tool were broadly described in [7]. Additionally, Grasshopper offers a mesh-based physics simulation tool called Kangaroo [34], through which form-finding for membrane structures is also available. Further details on form-finding with regard to a combination of both physical and computational approaches can be found in [35,36]. Numerous patterning techniques are available regarding the developability of textile materials and compensation of material deformation due to discretization [37] stresses and material properties [38]. A review on selected pattering techniques was presented in [39]. Further information on patterning methods can be found in [40,41]. A more advanced method of form-finding that preconceives eventual patterning with regard to its developability was described in [42]. Although some factors related to material behavior were omitted during our design process due to a simplified approach, Kiwi!3D allowed us to complete the whole digital design process, from concept to production drawings, within a single CAD environment and without losing the quality of the model, i.e., remeshing of NURBS surfaces. Such a simplified approach is still valid for temporary objects that are geometrically correct and eye-catching. The design and the construction of such temporary structures are today within the reach of designers thanks to the increasing accessibility of computer-aided engineering tools such as Kiwi!3D.

### 5.2.1. Form-Finding

Form-finding strategies have been developed ever since they were first used on a large scale to intentionally devise an optimal form from a structural point of view. Examples of well-recorded and documented cases of early form-finding techniques used in architecture and engineering are the catenary model of Sagrada Familia by Antoni Gaudi and Ponte Musmeci (Musmeci Bridge) in Potenza by Sergio Musmeci [43]. Physical form-finding methods, although powerful and allowing to optimize a structure's form while gaining greater spans, heights, etc., are limited by the natural behavior of materials used for simulations (e.g., strings, rubber bands, springs, soap films). Today, the awareness of the feasibility of unconventional structures is growing. So-called free-formed structures, released from the limitation of orthogonal forms, have opened a wide range of studies focused on revealing the scope within which these structures can be prefabricated and

simplified. For example, double-curved glazed grid shells are composed of parts that are not individually curved (rods, glass panels) [44], and membrane structures are composed of patches that are flat in their original configuration, such that, as a resource, they can be manufactured in rolls of long, developable bands.

This is called fabrication-aware design, a term introduced to emphasize the responsibility of designing nonorthogonal structural and architectural systems [45]. In addition to answering what shape would be optimal for a particular system of supports and edges of a membrane, it is important to include fabrication factors into the process of form-finding. In the case of membrane structures, these include the anisotropy of materials (fabrics tend to deform more in directions diagonal to their fibers), initial raw material dimensions, and different properties of material along seams. These and other purely geometrical factors are abstract when form-finding membrane structures are used with soap films. However, digital simulation tools allow us to take some of these abstract factors into consideration [32].

The profound understanding of membrane structures is owed to Pritzker Prize laureate Frei Otto, an architect who has been studying these structures morphologically [46]. His first structures (from 1955) were form-found with the use of physical methods such as soap films and stockings. Otto's membrane structures gained worldwide attention on the occasion of the World Fair in Montreal 1967. The Germany Pavilion, covered with tensile canopies, promised technological advancements in terms of light weight, versatility, and durability. In 1969, Frei Otto was invited to cooperate with a design studio which won the competition for the Olympiastadion in Munich. The winning project was inspired by Otto's membrane structures. In this case, the tensile structure had to permanently cover about 88,000 square meters. Although the former approach of iterative form-finding and building a physical model in a trial-and-error manner was sufficient, for the purpose of Olympiastadion canopies, CAD modeling was also introduced for their design.

In order to design a shape for the membrane, Otto used elastic stockings or soap films spread across the preconceived edge borders. The second method was more accurate despite being limited to a small scale. Tensile forces in stockings are disrupted by forces caused by elastic deformations, whereas, in soap films, such disrupting forces are eliminated (despite gravitational forces). Studying soap films allows finding forms (hence, form-finding) of thin membranes which are solely under tensile forces. When in equilibrium, tensile forces are tangent to the surface of a film. At each point of a membrane, two pairs of tension force vectors can be derived, which cancel each other out.

The two opposite vectors are not quite collinear due to the curvature of the surface, and their resultant vector is perpendicular to the membrane at a selected point; this force tries to move the membrane perpendicularly. However, this perpendicular force is cancelled out by an opposite force, resultant from two opposite vectors of tangent forces that act in a perpendicular direction. Hence, if no other forces act on a membrane (e.g., pressure as in pneumatic membrane structures), in order to obtain equilibrium, the curvature of the two perpendicular directions have to be opposite, i.e., the surface has to be anticlastic, with a negative Gaussian curvature sign.

This principle is naturally obtained by soap films, which, at each of its points, is moved toward the prevalent perpendicular force. This happens until the film takes a form in which both opposing perpendicular forces at each point of that form cancel each other out, i.e., an equilibrium is reached. Tensile forces are transferred onto edges which are static or onto pretensioned cables. Extrinsic form-finding is explained in the paraphrased quote below.

"Form-finding is a physical setup where a form self-organizes, and it is not drawn by hand, invented, or preconceived. It emerges in a physical process."

Patrik Schumacher [47]

A greater curvature leads to greater perpendicular forces. If the individual curvatures along specific directions are not equal, the forces are balanced with tensile stresses; hence, tensile forces are usually greater along directions with lesser curvature and have to be

compensated for by prestressing (the differences in forces as a function of the direction are shown in Figure 8). This can be done by material reduction along a specific direction (the force is obtained from the elastic deformation of a membrane) and by adding tensioners along specific edges.

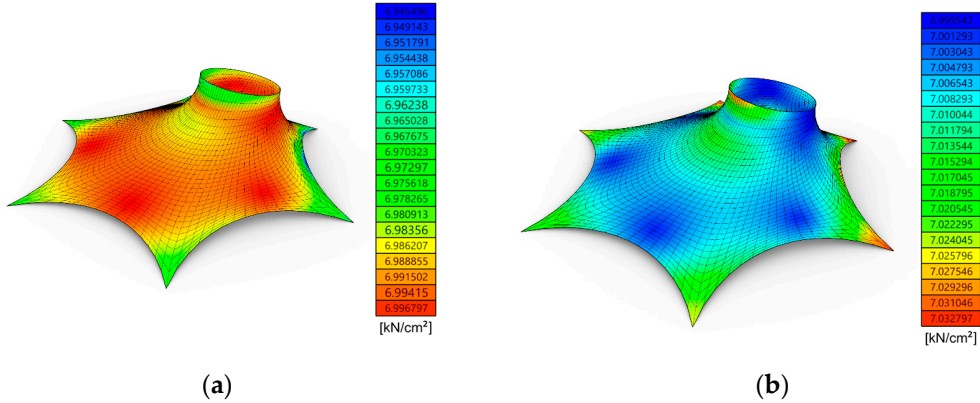

(**a**)　　　　　　　　　　　　　　　　(**b**)

**Figure 8.** Analysis of tensile forces in form-found membrane in the (**a**) latitudinal and (**b**) longitudinal directions. Differences in tensile forces occur as a function of the directions. Both stocking- and soap-film-based form-finding methods are burdened with built-in errors (elastic deformation forces and scale limits), which also overlap with the digitalization issue. Although very useful for educational reasons, geometrical forms of soap bubbles and membranes are not obvious for everyone; digital form-finding allows us to instantaneously overcome the obstacle of digitalization.

### 5.2.2. Form-Finding of the Outer Skin of the Obverse/Reverse Pavilion

For our case, we implemented two approaches using the Kangaroo 3D addon for Rhino3D (mesh based, interactive physics/constraint) and Kiwi3D Iso-Geometric Analysis tool in the Grasshopper environment. The results are shown in Figure 9.

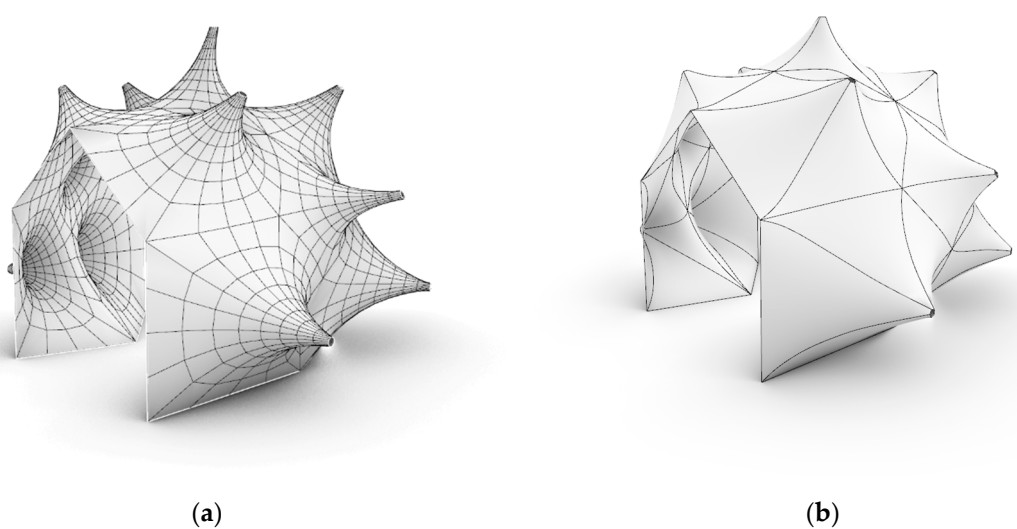

(**a**)　　　　　　　　　　　　　　　　(**b**)

**Figure 9.** Comparison of two models: (**a**) minimizing edge lengths of mesh; (**b**) isogeometric analysis of NURBS surfaces.

In Kangaroo 3D, a form is obtained according to a setup of geometrical goals. In the case of membranes, an edge length goal is usually used, which constrains the Kangaroo solver to find a setup of vertices (of a mesh) that results in the smallest (for the sake of membranes) value of all distances between connected vertices. The target length of an edge can be set to 0, any other factor or initial length, e.g., 0.5 or 1.5, or any other positive real number. However, lengths between corresponding vertices do not correspond with the

tensile forces that would appear in a soap film. This approach is burdened with a built-in problem, i.e., the final state highly depends on the initial shape and discretization (topology of the analysis mesh).

Meshing (discretization), by means of finite analysis methods used for form-finding of membrane structures, should not be confused with meshes used for physical/geometrical constraint solvers. In the first case (FEM), each pair of adjacent facets of a mesh (for R2 surfaces) is described by simple functions regarding their interaction and assuming their states. A detailed comparison between Kangaroo 3D and natural force density methods was presented in [33].

The isogeometric analysis that we performed with the use of Kiwi!3D, i.e., the form-finding process, allowed us to bypass the discretization step. Shifts of analyzed patches of the membrane under imbalanced perpendicular counteracting forces were introduced into a NURBS model by means of its control point transformations. A deeper theoretical explanation was provided in [28].

As seen in the comparison (Figure 9), differences between the two approaches were clearly visible.

### 5.2.3. Creating Developable Patches

Although it may seem that full-scale membranes gain their shape due to elastic deformations of originally flat material sheets, their shape is, in fact, primarily achieved due to an assembly of material patches that are developable, and they are only secondarily achieved due to elastic deformations.

Any form of double-curved surface (such as the anticlastic surface of a membrane) cannot be obtained from a sheet of material that is originally flat, due to its holonomy, i.e., the fact that closed shapes around measured patches of surfaces have smaller (for synclastic) or greater (for anticlastic) internal angle sums than they would have if drawn on flat surfaces (see Figure 10).

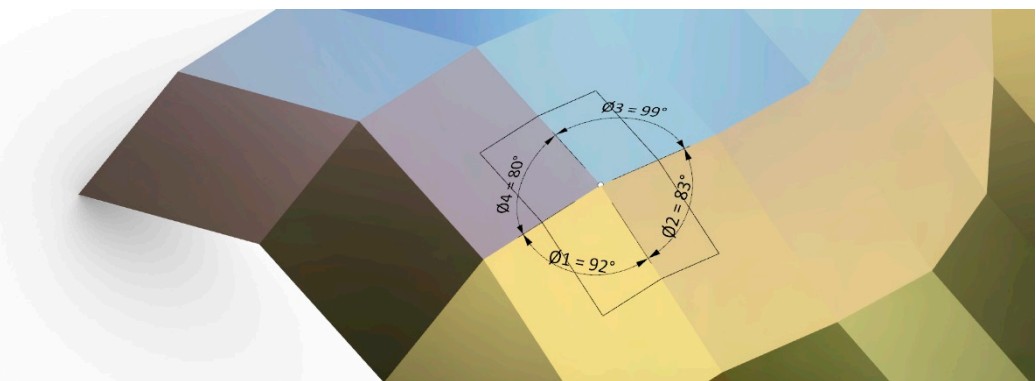

**Figure 10.** The total angle around a vertex is not a full angle on synclastic surfaces, whereas it is more than a full angle on anticlastic surfaces.

This means that the material is in deficit (anticlastic surfaces) or in excess (synclastic surfaces) of a full angle. To a certain extent, these differences can be compensated for by elastic deformations of flat material pieces caused by prestressing; however, on a global scale, these differences are compensated for by the assembly of flat patches. These flat patches can be curved directionally when connected together, thereby gaining the form of developable surfaces.

One of the main goals of designing membrane structures is finding or devising the shapes of these patches. In our case, factors related to the material's anisotropic structure, its imperfections (for large-scale membrane structures, a structure is assembled from patches fabricated from materials produced in a single production batch in order to avoid any differences), small elastic deformations within single patches, and prestressing along with lesser curvatures were omitted in order to simplify the process and obtain a result

sufficient for a temporary pavilion structure. The fabric, after assembling individual patches, was connected to the base structure without the use of tensioners to simplify the design. The resultant form of membranes was similar to the digital model obtained in the form-finding process; however, some acceptable imperfections were observed, such as wrinkling resulting from uneven stresses along opposite directions or excess material.

As input, IGA analysis takes flat, single-curved, or double-curved NURBS surfaces and plots a result composed of such surfaces. This resultant 3D model of the pavilion's external skin was then further processed in order to devise individual shapes of material patches. Initially, the surface of the membrane was divided into parts corresponding with each rod pushing it outward. Such parts were close to the forms of cones with polygonal bases and additional longitudinal curvatures. Then, each 'cone' was further divided around its axis along the longitudinal direction, thereby creating triangle-like constricting patches with concave edges. The divisions were made using geodesic curves, i.e., curves that trace the shortest path between two points on curved surfaces. Pairs of these division geodesic curves were then used for the definition of ruled, developable surfaces capable of unrolling on a flat surface, using their outlines as fabrication patterns. Through this simplification, any excess material between geodesics that protruded over the simplified developable surfaces was flattened. A greater distance between geodesics results in a greater simplification error; therefore, decisions regarding the number of divisions around each cone represented a tradeoff between precision and fabrication complexity.

A greater surface deficit between adjacent patches leads to a greater distance between opposite concave edges (Figure 11). When all patches were oriented on a flat surface, one more step was carried out, in which some of the adjacent pairs of 'triangle like' patches from different 'cones' were re-joined if the concavity of the opposite edges indicated a negligible surface deficit, thus allowing a reduction in the number of seams.

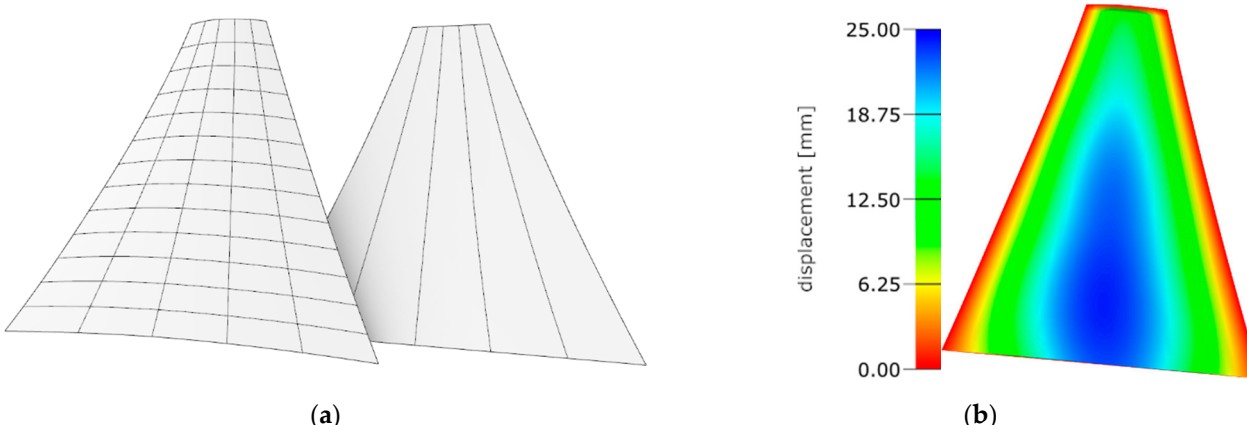

**Figure 11.** (**a**) Comparison of original double-curved patch and its simplified, ruled version. Curves indicate curvature directions; (**b**) colored map indicating the displacement between the double-curved and developable surfaces.

Although Rhino allows 'developing' any ruled surface, it must be taken into account that not every ruled surface is a developable one. A ruled surface defined by two opposing edge curves and a straight line 'sliding' along them is not considered developable if it has nonzero Gaussian curvature. For example, a hyperbolic paraboloid is a double-ruled surface with a negative Gaussian curvature, which cannot be 'flattened'; hence, it is not developable.

Intrinsically, a developable surface has to be composed of straight sections, among which those next to each other are also coplanar. Anticlastic ruled surfaces have straight sections that are skewed toward each other; hence, they are not developable. The above description refers to discrete models of rule surfaces that are represented by a finite number of straight sections. In terms of the continuous surface representations that we dealt with, a surface is defined by an infinite number of straight lines with infinitesimal distances

between neighboring ones. Instead of performing a differential analysis measuring distortions between corresponding sections, a Gaussian curvature analysis can be used, which is available as a built-in function in Rhino3D. In Rhino 3D, a continuously curved ruled surface is built as a set of infinite straight sections, each defined by two points on opposite directrix curves. Coordinates of 3D points defining straight sections are calculated for each directrix curve using their parametrical representation, i.e., a mathematical function which takes a parameter as an argument and plots a 3D point. Such a parametric curve is characterized by an interval of numbers, usually between 0 and 1. Corresponding points creating straight sections are calculated from parameters that are proportionally assigned between two directrix curves, in which case a developable surface is not necessarily obtained. Assigning corresponding parameters between directrix domains is a bijective function (excluding cases of NURBS surfaces with singularity points), which does not necessarily assign values of parameters proportionally. In Rhino3D, a user is allowed to manually input additional, arbitrarily chosen straight sections between directrix curves, thereby changing the mapping of parameters between curves. When manually adding straight sections, the user has to iteratively measure the Gaussian curvature of the created ruled surface until it is acceptably close to zero at every point on that surface.

Newer versions of Rhino3D offer a single function that allows creating developable ruled surfaces, whereby parameter domains of directrix curves are automatically mapped such that the corresponding generatrix lines are coplanar (by means of discrete geometry) or the Gaussian curvature of the surface is equal to zero at every point.

The final 3D model of the pavilion and patterning is shown in Figure 12.

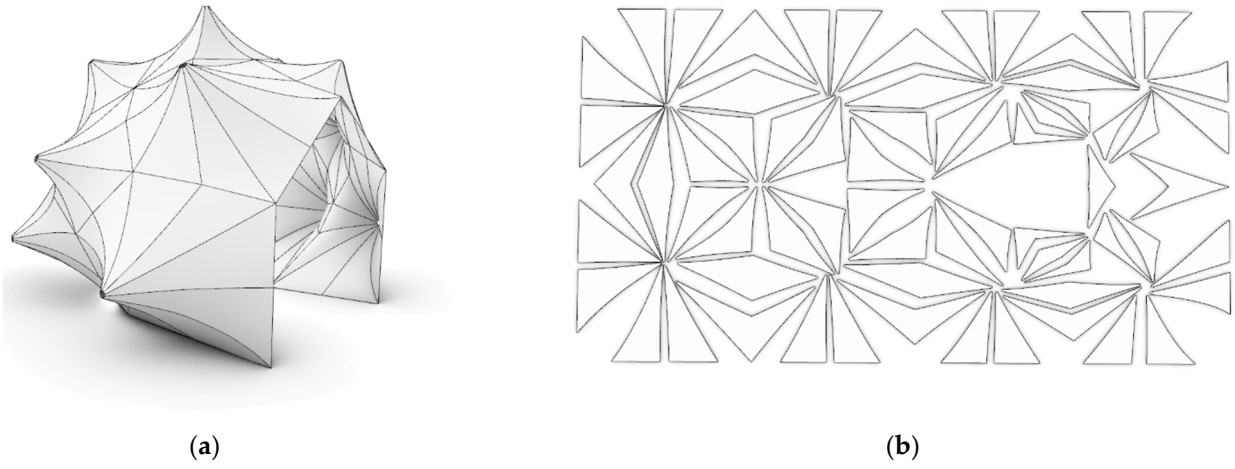

(**a**)                                   (**b**)

**Figure 12.** (**a**) the 3D model with simplified, ruled patches; (**b**) patches in a flat configuration, prepared for fabrication.

## 6. Results—Building the Pavilion

The Obverse/Reverse Pavilion was prefabricated at the ProtoLAB Laboratory at Wroclaw University of Science and Technology in cooperation with the students of architecture.

The main structure of the pavilion consisted of five frames (Figure 13). There were two pairs of larger frames (type A and type B) and one middle frame (type C). The sizes of the frames varied from 290 cm × 300 cm to 243 cm × 252 cm.

Each of the frames was composed of four layers of 5 cm thick cardboard honeycomb panels strengthened by lamination with four layers of timber-based 10 mm OSB (oriented strand board). The sandwich element had a thickness of 24 cm and was 20 cm wide. It was composed of 83% paper by volume; however, it retained its expected structural stiffness.

For transportation purposes, the frames were prefabricated in halves. The ends of the halves overlapped and were fixed to each other at the building site with screws. The frames were connected to each other by means of 18 mm OSB floor plates, 32 4 × 4 cm wooden battens in the pavilion's longitudinal direction, and eight 15 × 1 cm planks fixed diagonally to the battens to stiffen the structure against lateral forces.

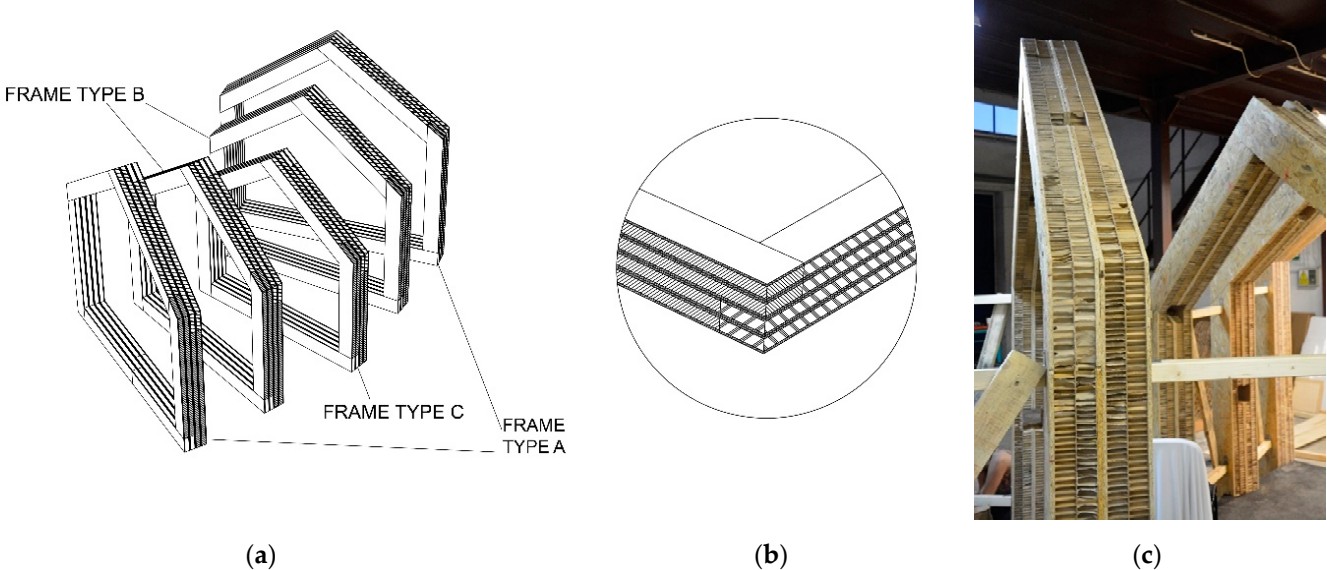

|(**a**)|(**b**)|(**c**)|

**Figure 13.** (**a**) Substructure frames; (**b**) connection between frame halves; (**c**) sandwich frame composition.

The membrane, due to budget constraints, technical parameters, and the expected life span of the structure, was made out of Codura, which is a polyester textile additionally impregnated with a PVC layer. The planar patches presented in Figure 12b were cut from the roll with 1.5 m width and 80 m length. Each of the patches was cut with an extra 1.5 cm margin for sewing.

The membrane was sewn with the use of folded seams. First, two pieces of material were sewn with a normal stich, and then the material was folded in a zig-zag pattern and sewn with cross-stich. Such a solution minimized ruptures caused by tensile forces by transferring the forces from the stich to the material strength.

The largest forces on the membrane were concentrated at the peaks. To ensure freedom of movement, solid fixation, protection from rain, and the possibility of wet air evaporation, the cross-belts were sewn on the peaks and covered with textile caps that allowed convective movement of the air (Figure 14).

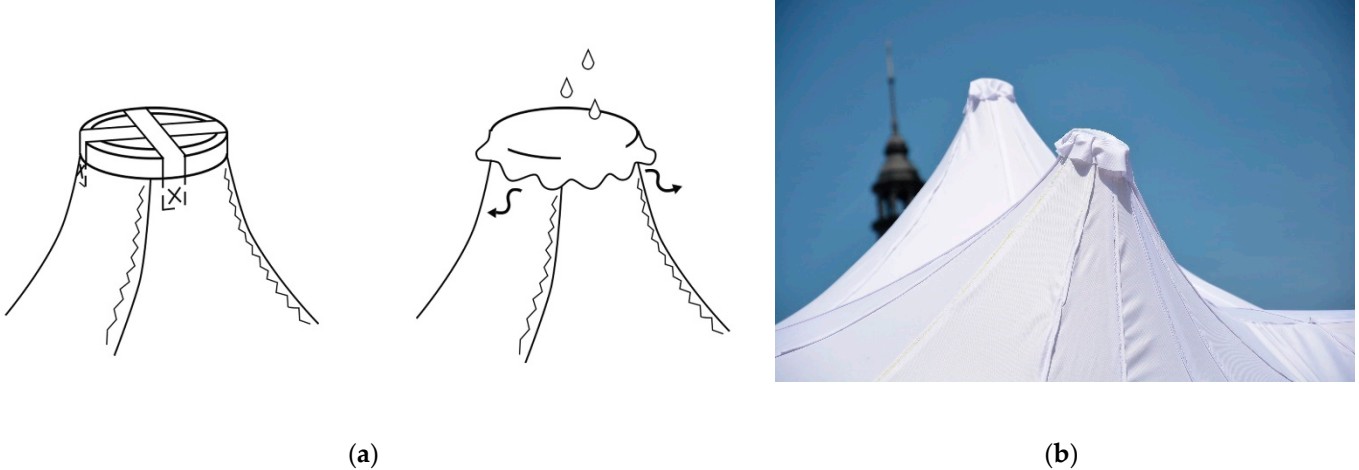

(**a**)　　　　　　　　　　　　　　　　　　　　　　　　　　　(**b**)

**Figure 14.** (**a**) Ends of the outer membrane peaks; (**b**) outer membrane peaks.

The finished pavilion was 485 cm long, 550 cm wide, and 420 cm high, and its cost was slightly over 1000 EUR (excluding transportation and production; see Table 1).

**Table 1.** The Obverse/Reverse Pavilion budget. The presented costs do not include transportation and human labor. The pavilion can be considered low-cost as the structure with an area of 26.6 m$^2$ had a budget slightly higher than 1000 EUR.

| Product | Amount | Total Price (EUR) |
| --- | --- | --- |
| OSB (1250 × 2500 × 10 mm) | 20 | 250 |
| OSB (1250 × 2500 × 18 mm) | 3 | 57 |
| Plywood (1250 × 2500 × 4 mm) | 2 | 8 |
| Timber planks and boards | 50 m | 66 |
| Codura textile | 80 m | 263 |
| Honeycomb panels (1250 × 2200 × 50 mm) | 20 | 87 |
| Glue | 30 L | 46 |
| Polycarbonate plate | 1 | 86 |
| Paint | 2 L | 13 |
| Miscellaneous (brushes, hinges, screws, etc.) | - | 30 |
| Lightening | - | 66 |
| Production materials (mock-up + tests) | - | 110 |
| Paper Tubes (100 × 10 × 2500 mm) [1] | - | - |
| Total | | 1082 |

[1] Paper tubes were sponsored by Corex.

*Assembly*

After prefabrication, the pavilion was transported to the site, i.e., Plac Wolności (Square of Freedom) in Opole. First, the sandwich cardboard/timber frame halves were assembled. Subsequently, the frames were connected with floor panels, horizontal beams, and diagonal planks (Figure 15). The frames were covered from the outside with self-adhesive PVC foil to prevent material moisture. After completion of the main structure, the outer skin membrane was placed on top of the pavilion.

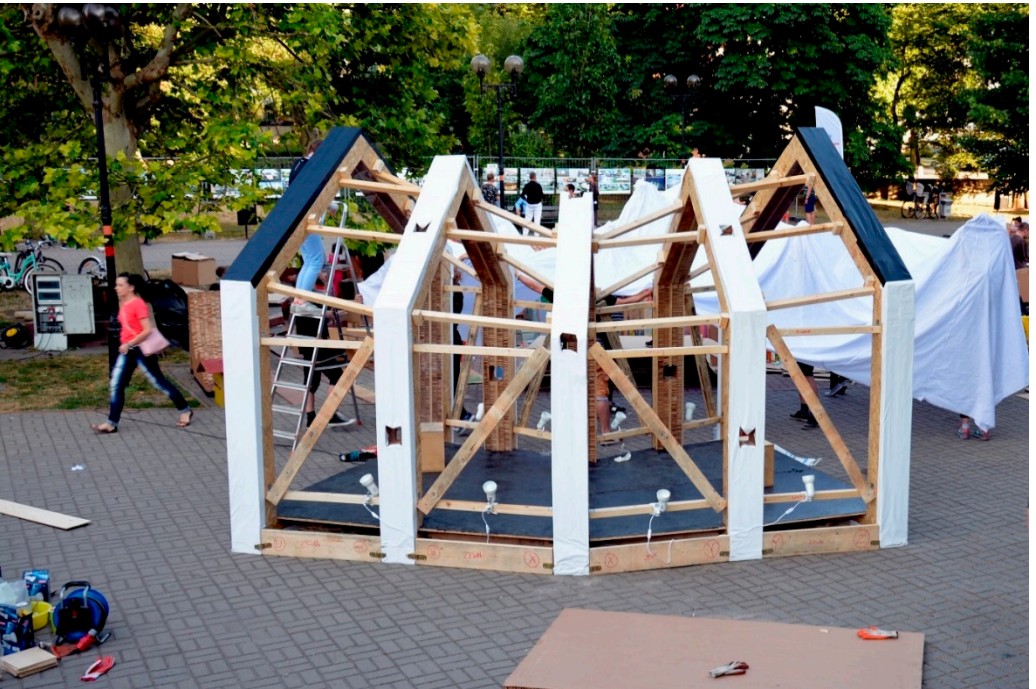

**Figure 15.** The substructure assembly process.

Next, 11 paper tubes with a length of 250 cm, inner diameter of 10 cm, and wall thickness of 1 cm thickness were inserted into the holes in the cardboard/timber frames and pushed outward in order to stretch and position the outer membrane (Figure 16). The tubes were fixed to the frames with bolts, and the parts that protruded from the inner side

of the frames were cut (Figure 17a). The membrane was fixed to the bottom and both ends of the pavilion with nails, and it was additionally covered with 4 mm plywood along the front and back entrances (Figure 17b). Finally, the light installations and inner carboard walls were fixed. The inner walls were made of corrugated cardboard plates with black self-adhesive PVC foil on both sides.

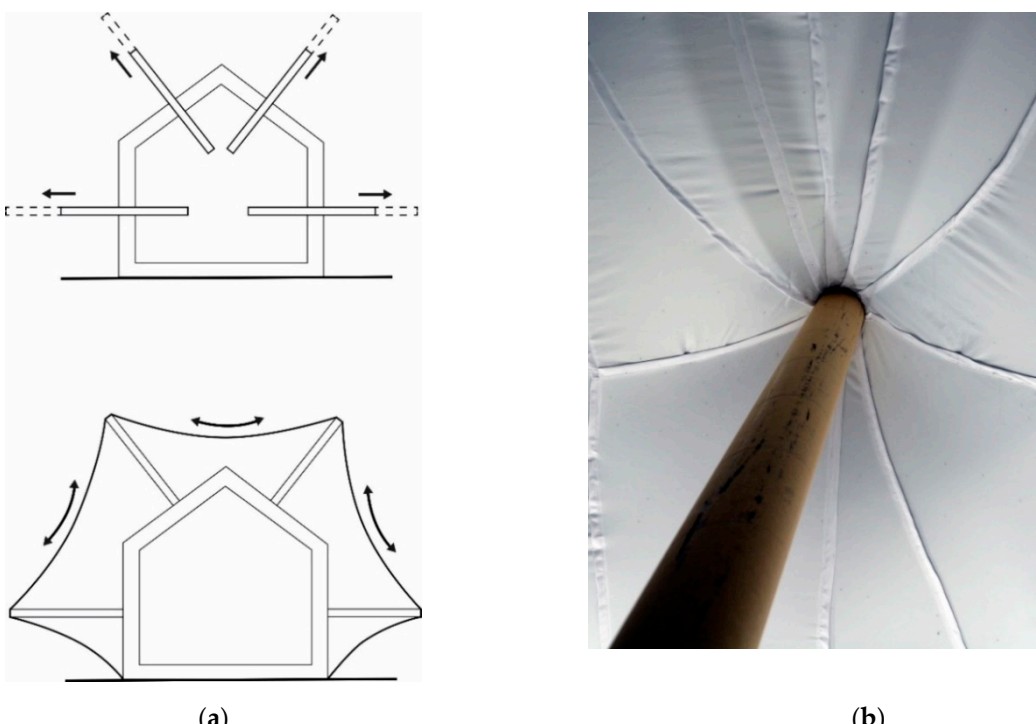

(**a**)                                                                              (**b**)

**Figure 16.** Stretching the outer skin membrane: (**a**) diagram illustrating the stretching of the membrane with paper tubes; (**b**) paper rod used to stretch the membrane.

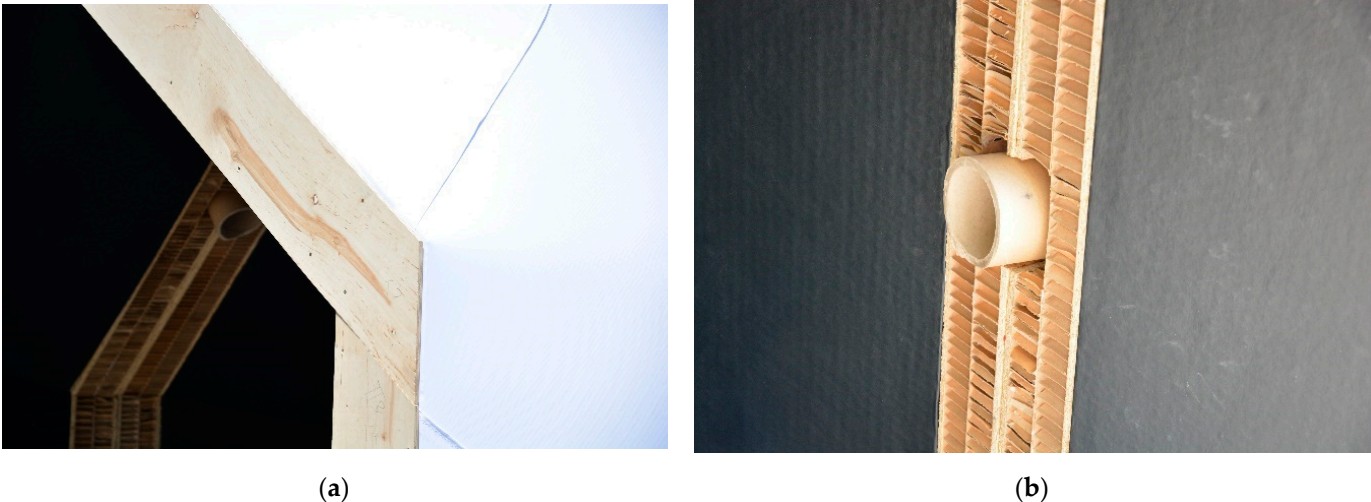

(**a**)                                                                              (**b**)

**Figure 17.** The pavilion details: (**a**) plywood clamp holding the membrane at the end of the pavilion; (**b**) fixation of paper tube in the substructure.

The Obverse/Reverse Pavilion was exhibited at Plac Wolności in Opole for 3 days (Figures 18 and 19). It was accompanied by cardboard chairs and benches designed especially for this occasion.

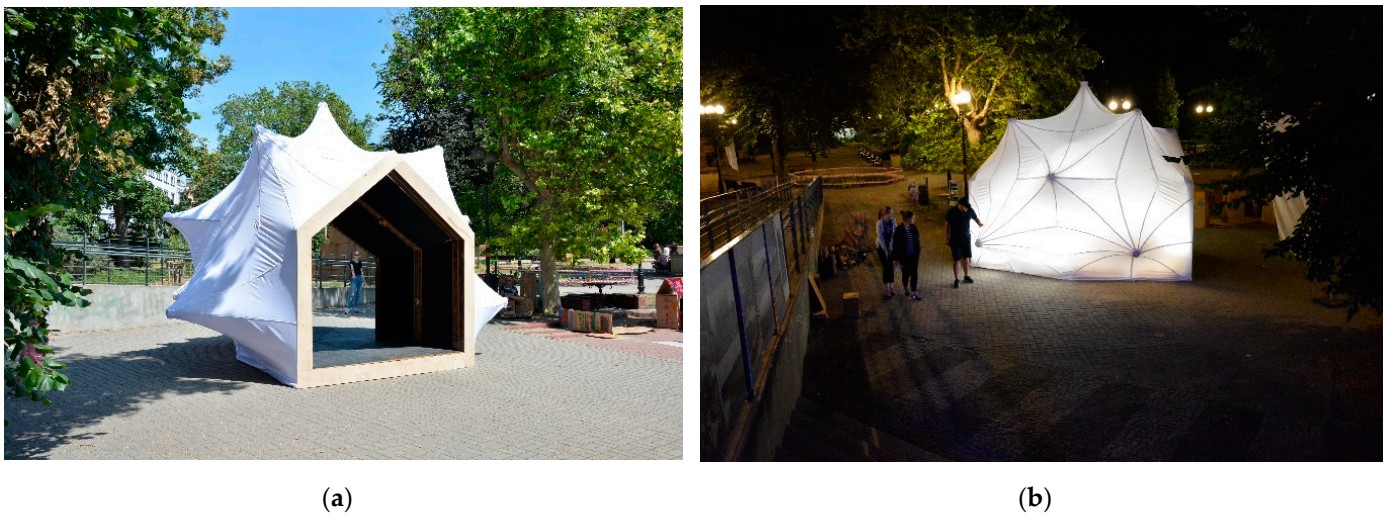

(**a**)                                                      (**b**)

**Figure 18.** The Obverse/Reverse Pavilion: (**a**) pavilion in daylight; (**b**) pavilion at night.

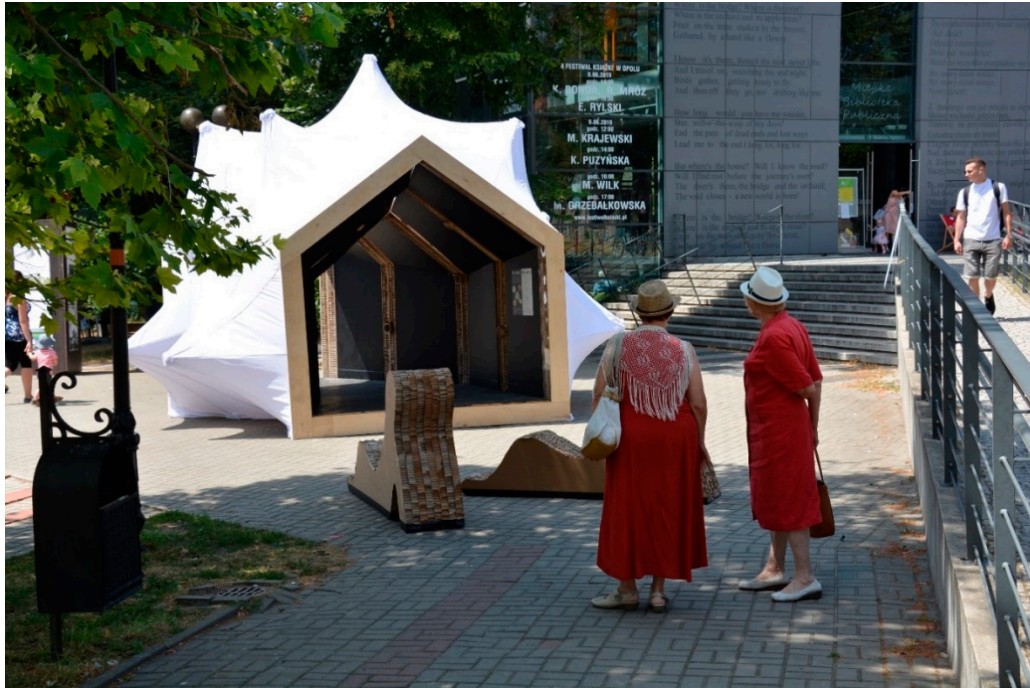

**Figure 19.** Obverse/Reverse Pavilion in the public space of Plac Wolności.

The pavilion was of considerable interest to visitors. Its surprising form attracted attention; as the shape of the structure was not exactly obvious, passers-by were eager to come closer, go through, and even touch the outer skin.

## 7. Discussion

The example of the Obverse/Reverse Pavilion shows how even small pop-up structures in an urban area can bring attention and create new positive value. The pavilion, due to its unconventional form and used materials, became a temporary sculpture. It was erected in 1 day, and it was exposed in its primary location for just 3 days. The temporary characteristic of the structure and the assembly process enhanced public attention, thus piquing the interest of citizens and tourists attending other events on the International Day of Architecture.

Previous pavilions attracted the interest of passers-by, albeit in a different manner. The pavilions from the years 2016–2018 were made out of straight timber elements. Their

structural system was easy to understand as they were composed of diagonal (the Architectonic Sculpture) or vertical and horizontal beams (Wood and Shadow Pavilion), or plates (Water Pavilion). On the other hand, the Obverse/Reverse pavilion was perceived as an incomprehensible, weird form. The structural system, as it was hidden between the straight planar elements of the inner shape and the outer blob-like skin, was impossible to understand. Thanks to its parametric design, the skin achieved an unobvious shape. The soft and friendly form encouraged visitors to come closer and walk through, and, similarly to the Water Pavilion, interactions with this urban sculpture were welcomed. The behavior of passers-by who went through the pavilion, walked around, and touched its outer skin led the authors to conclude that the pavilion fulfilled its role as a pushpin in the urban fabric of a city, corresponding with the visual and haptic sensory systems.

However, the strongest point of the pavilion was its meaningful form. It told the story of the profession of an architect.

The other pavilions were built for a short period of time (maximum 1 week). The Obverse/ Reverse pavilion was exposed for a total for 6 months. This required special treatment, especially considering the fact that it was made of materials susceptible to humidity and water. Covering the structural frames from the outside with self-adhesive PVC foil provided the necessary protection from the ground, whereas the outer skin membrane made of watertight polyester with a PVC film protected the pavilion from the rain.

Its limited color palette, i.e., black, white, and natural colors of timber and cardboard, were in a contrast to the overwhelming amount of information that needs to be absorbed in a city center. Unlike the previous pavilion which featured colorful lights, the Obverse/Reverse Pavilion fulfilled its purpose through its shape. The only information was a 50 × 50 cm poster hung on the inner wall, which explained the project idea and used materials.

The material choice was also something new. Paper-based elements are not perceived as durable building products. Exposing them as structural elements, in comparison to the earlier pavilions made of solid timber elements, gave the Obverse/Reverse Pavilion an impression of delicacy.

The structural system, composed of structural frames and an outer membrane skin stretched on rods, allowed for the creation of volume through the use of voids. Accordingly, the amount of material and, thus, the budget were minimized.

In general, the design and production of the Obverse/Reverse Pavilion was a success and it met the expectations of the client, i.e., the Chamber of Polish Architects branch in Opole. The design goals set at the beginning of the process were fulfilled, i.e., a meaningful and eye-catching form, influence of the visual and haptic sensory system, an unusual shape that created some kind of narration, and an inclusive form.

However, several elements could be improved or further researched. The structural system could be better thought through in order to make it easy for transportation to the second location. The pavilion had to be dismantled and rebuilt next to the Contemporary Art Gallery. The new system could allow for harmonica-like folding of the structure, where all the frames would be placed next to each other, whereby the membrane would not have to be detached. Surprisingly, the biggest problem in the production phase was caused by cutting the horizontal 4 × 4 cm beams. As they were attached to the main structural frames at two different angles (horizontal and vertical) from each side, there was a misconception when the dimensions were read from the software. A better marking of the elements and projection of the drawing planar surface would allow for more accurate cutting.

From the material point of view, deeper research could include the material properties, especially the strength tests on the paper tubes and the sandwich panels. The paper tubes seemed to be too thick; thus, their weight and the amount of material could be minimized. The structural frames could be made of honeycomb panels sandwiched with thick full cardboard. Such a solution would allow using as much paper-based material as possible. The outer skin could be made from a more eco-friendly material. Another feature required for such a skin would be an elastic material. This, however, could increase its cost or make

it more susceptible to moisture. Therefore, an additional impregnation layer would have to be applied on the final pavilion.

In order to minimize material costs for a pavilion, a nesting technique can be applied. Free-formed objects tend to be composed of parts that are irregular in length, shape, and geometry [44]. In the case of membrane structures, these irregular parts are flat material patterns that differ in shape. These patterns have to be cut from a single band of textile with constrained dimensions. Naturally, minimization of waste material is desirable. Nesting is the task of arranging pattern outlines over the shape of a raw textile outline in such a manner that they occupy the smallest possible area, thus minimizing the waste material. The rotation of patterns is also possible. Such a task is a combinatorial optimization problem that can also be performed computationally in Grasshopper.

From a general perspective, due to the broad development of form-finding tools and patterning techniques available for nonprofessionals in the engineering field, a comprehensive overview is required to highlight possible drawbacks and potential obstacles when using of such tools. Such a review can also help the architect to decide whether a professional approach is required in order to design and build a long-standing structure or whether such aid is not necessary, e.g., in the case of low-cost temporary structures.

Due to the use of computer representations of the project in every phase from conceptual design through to production, it was possible to foresee potential mistakes, e.g., incorrect connections or structural element parameters. This also allowed the authors to develop a three-dimensional tensile membrane which could be divided into flat patches and was relatively easy to produce.

Even though modern membrane structures have been known and used in architecture and civil engineering for over half a century, there are still a wide range of applications not yet explored. Rapidly evolving design tools and methodologies are becoming more ergonomic and accessible from a designer's point of view, thereby allowing them to emphasize the creative aspects of their work. New, morphologically complex membrane structures can be form-found, which was not possible before using natural form-finding processes [48].

Membrane structures are also being explored toward finding new applications, e.g., as lost form/scaffolding and reinforcement for shell structures and as formworks of free-formed concrete structures, which impose a necessity for the production of otherwise expensive and material-consuming formworks [49].

Parametric design tools, as used by the authors, as well as all other tools that allow for an easy design of complex structures, currently play an important role in education, complementing the understanding of material and structure behavior, previously taught to architectural students through physical contact and experimentation [50]. However, the improper usage of these tools without an essential background and understanding of particular material properties or structural systems may lead to unexpected and dangerous results. The dark side of the "Bilbao effect" is represented by the trend of so-called "blob architecture", which was quickly criticized for not meeting the aesthetic, durability, or predictability expectations and, therefore, fell into recession [51].

It is also worth noting the importance of model building and form-finding through both physical and digital models, as also highlighted by previously cited researchers [36]. A digital model cannot fully characterize material behavior, and pictures of membrane structures do not allow an understanding of the formation of minimal surfaces, such as experiments with soap bubbles. Hence, the Obverse/Reverse Pavilion also performed a didactic function, since it was built by designers who had direct experience with the behavior of the membrane.

The construction of the Obverse/Reverse Pavilion allowed us to address the above-mentioned issues and embed a mature approach among the students who participated in the design and construction process.

## 8. Conclusions

The contemporary world is characterized by the immensity of events that most often have a temporary nature. An enormous amount of information is sent to users of public spaces. This information has to be absorbed, processed, and synthesized, after which the user can decide on its importance and relevance. Architects and artists, especially when working with temporary structures, search for new unconventional materials and formal solutions in order to win the battle for citizens' attention. The use of computer-aided architectural design, especially in terms of its parametric potential, allows for the creation and optimization of unexpected free forms.

Abstract form-finding methods are now undergoing their renaissance, and using them frequently opens the way for world-class architecture. When studying the background of modern Pritzker Prize laureates, it becomes clear that the pursuit of innovative methods for obtaining structural and material efficiency is playing an increasingly important role, particularly in an era of growing awareness with respect to resource scarcity and environmental conditions. A special example of cooperation between engineers and architects that resulted in a structure with emergent quality was that with structural engineer Cecil Balmond, whose design philosophy is based on abstract factors, mathematics, emergence, and fractals, and for whom a structure as conceptual rigor is considered architecture. The strength of such an approach is emphasized by the fact that this structural engineer cooperated with many Pritzker Award winners (Toyo Ito, Daniel Libeskind, Álvaro Siza Vieira, Eduardo Souto de Moura, Rem Koolhaas, Shigeru Ban (Centre Pompidou-Metz), and James Stirling), producing structures of unprecedented value.

Most novel structural systems such as taut structures (membranes, pneumatic structures, and cable structures), bending-active structures, reciprocal frame structures, or shells, which are less common in permanent architecture attached to orthogonality, promise very good characteristics in terms of cost and material consumption for the span/volume of such structures. They are also usually convenient for prefabrication and quick assembly. On the other hand, they require a deep understanding of the theoretical background of their design, since, in form-found structures, it is not the designer who decides on their form, but the structures themselves.

The Obverse/Reverse Pavilion was an example of a surprising and organic form made of simple, low-cost, and pro-ecological materials such as timber- and paper-based elements, and a polyethylene membrane.

The timber-based elements were OSBs (oriented standard boards), plywood boards, wooden battens, and planks. OSB and plywood are timber-based materials and are biodegradable. During their production, a phenol resin is used for gluing the wood shavings or veneer. The product can be recycled and used for chipboard production, or it can be chipped and used for landscaping or for generating energy and electricity. OSB is regarded as more eco-friendly than plywood as it is made of wood shavings glued together with resins; therefore, it can be made using small trees [52]. In Europe and the United States, the raw material for plywood and OSB is obtained from certified forestry. In general, timber-based building products are considered more sustainable than traditional products such as concrete and steel [53].

The paper elements used for the pavilion production were paper tubes and honeycomb cardboard panels. Paper tubes and honeycomb panels were produced with both kraft paper (made of virgin fibers) and testliner (made of recycled fibers). Both products can be recycled. Paper can be recycled up to seven times; however, each time, new fibers are added to the batch [27]. The recycling of paper tubes is harder, as they consist of many layers of glue; however, it can take place in special plants or at paper-tube production facilities. Honeycomb panels are easy to recycle and can be disposed of in common bins. The structural frames were made out of OSBs and honeycomb panels, where 83% of the structure by volume constituted honeycomb panels.

The outer skin of the pavilion was made of Codura, a strong waterproof textile with a layer of PVC. It is mainly used for bags, tents, and sun- and rain-protective devices

such as umbrellas and awnings. Codura is made of polyester. Polyester is manufactured from mineral oil and it is not biodegradable; additionally, its production consumes a large amount of energy. On the other hand, polyester is a recyclable material, and the ecological footprint, represented in global hectares (gha), of producing one metric ton of spun fiber is lesser than that used for hemp or organic cotton production. This is due to the fact that there is no land use necessary for polyester cultivation [54].

The above information allows us to conclude that the Obverse/Reverse Pavilion was an eco-friendly structure.

The formal effect was achieved by the use of a parametrically designed outer skin in the form of a minimal surface, stretched over the substructure.

Membranes are suitable for temporary objects thanks to their light weight and low production cost; however, the design and the implementation of membrane structures require additional skills and knowledge of the designers.

The implementation of parametric design methods allows achieving a spectacular form of an architectural object despite a limited budget and tight schedule.

**Author Contributions:** Conceptualization, J.F.Ł.; methodology, J.F.Ł., M.Ś.; software, M.Ś.; investigation, J.F.Ł., M.Ś.; resources J.F.Ł., M.Ś.; writing—original draft preparation, J.F.Ł., M.Ś.; writing—review and editing, J.F.Ł., M.Ś.; project administration, J.F.Ł.; funding acquisition, J.F.Ł. All authors have read and agreed to the published version of the manuscript.

**Funding:** This research was co-funded by the Chamber of Polish Architects branch in Opole. The paper tubes were sponsored by Corex.

**Data Availability Statement:** The data presented in this study are available on request from the corresponding author.

**Acknowledgments:** The authors would like to express their gratitude to all the students involved in the realisation of the Obverse/Revers Pavilion. Especially to: Weronika Abramczyk, Kiryl Furmanchuk, Orest Savytskyi, and Martyna Apczyńska, Klaudia Bociek, Aleksandra Jodłowska, Magdalena Jabłońska, Julia Kochańska, Kacper Kostrzewa, Kinga Niewczas, Michał Sobol, Gab-ryjela Szczerba, Kinga Wasilewska, Natalia Wojtaś, Dominika Zdaniewicz. We are grateful for the support of Philipp Längst, Anna M. Bauer, Alexander Michalski, Julian Lienhard from Ki-wi!3D IsoGeometric analysis tool, and the support of Romuald Tarczewski and Agata Jasiołek from Wroclaw University of Science and Technology. We also would like to thank to the mem-bers of Chamber of Architects, especially to Jakub Tomiczek, Janusz Śliwka and Kamila Wilk.

**Conflicts of Interest:** The authors declare no conflict of interest.

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
