# Peer review of "The Obverse/Reverse Pavilion: An Example of a Form-Finding Design of Temporary, Low-Cost, and Eco-Friendly Structure"

_buildings, doi:10.3390/buildings11060226_

Round 1

Reviewer 1 Report

Dear authors,

I thank you very much for this reading. I am an engineer, and an architects. I think you really described this issue. Your paper is really well-written, explains everything very well and it does improve the knowledge in the field. Congratulations

Reviewer 2 Report

The presented manuscript deals with the topic of design of temporary, low cost and eco-friendly structure. The article is very interesting but requires some minor corrections.

There is no reference to objects of a similar scale which were built earlier.

Furthermore, I would change the title, in order to be more pertinent to the content of the manuscript. In particular, the word "analysis" creates the expectation to see some numbers in the following. A lot of information is provided in the manuscript as well authors' own thoughts that are not in order. Chapters 2 and 3 are too extensive and unbalances the text.  The beginning of Chapter 3 should be entered in Chapter 2.

Ancient stone theater can hardly be seen as a temporary structure (Lines: 127-128).

There is an incorrect numbering of chapters, chapters 1-5 are followed by subsections 3.1, 3.2, 3.2.1, 3.2.2, again 3.2.2, and then 4-6.

Figure 6 is not innovative and is not necessary. On the other hand, Figure 7 should contain the legend and numeric values. The same with Figure 10.

The manuscript is characterized by a lack of proper discussion. The discussed structure should be compared with other structures of this type on a similar scale in the broadest context. For example with the Polish examples presented earlier. Authors should discuss the results and how they can be interpreted in perspective of previous studies and of the working hypotheses. Future research directions may also be mentioned.

The conclusions should present a scientific argument based on real data on the low cost of this type of structure and its eco-friendly features.

Reviewer 3 Report

This paper describes/reports a temporary pavilion project, claiming it is/was novel, attractive, low-cost, and eco-friendly design. The design is a bit interesting to me, but I am not sure if it is novel and attractive to the reader of this Journal. The text is well written and potentially contributes to architectural and design practice. However, the paper itself is hardly regarded as a research paper.  

A research paper consists of a clear research gap, question or problem, aim, and methodology that result in corresponding findings.  However, this paper does not properly describe these important components. Even if this paper was written for practice-based research, it still needs an acceptable literature review, aim, problem analysis and methodology.

Furthermore, it has a very limited contribution to this research community. Some sections describing the use of a parametric modelling tool for form-finding may be revised to be a research paper, but the application and its form-finding process should be properly analysed and verified. In order to do this, the authors should extensively review exiting studies on parametric form-finding. There is also an error in the section titles, 5 followed by 3.2…3.2.2 and then back to 4. I suggest the authors to develop this paper as a research paper in a conventional manner, or just find a place in a more practical architectural journal or magazine.

Round 2

Reviewer 3 Report

This revision is much better than before.

Well done in such a short timeframe.

There are two more suggestions.

One concern is there are multiple sub-questions that may be unnecessary. Are you clearly answering all in the manuscript? otherwise, please delete or reduce them.

The other suggestion is, if possible, describing a clear gap or limitation about previous computational methods using parametric design and how you have improved that in this project.
